Fish habitat restoration on the basis of water morphology simulation

Chen Xiaolong chenxiaolong@fmiri.ac.cn
Che Xuan chexuan@fmiri.ac.cn
Liu Xingguo
Zhu Lin
Tian Changfeng
Li Xinfeng
Fishery Machinery and Instrument Research Institute, Chinese Academy of Fishery Sciences , Shanghai , China
Meraj Gowhar
Electronic publication date: 2022 Aug 23
Publication date: 2022
Volume: 10
Electronic Location ID: e13943
Received 2022 Mar 28; Accepted 2022 Aug 3
Copyright: © 2022 Chen et al.
Copyright year: 2022
Copyright holder: Chen et al.
License: This is an open access article distributed under the terms of the Creative Commons Attribution License, which permits unrestricted use, distribution, reproduction and adaptation in any medium and for any purpose provided that it is properly attributed. For attribution, the original author(s), title, publication source (PeerJ) and either DOI or URL of the article must be cited.
License URL: https://creativecommons.org/licenses/by/4.0/

Keywords: Fish habitat, Restoration project, Diversion dam, Water morphological characteristics, Flow pattern simulation

Funding: Central Public-interest Scientific Institution Basal Research Fund FMIRI of CAFS 2020YJS001 Yangtze River basin ecological restoration engineering technology center of the Ministry of Agriculture 2021CG009 Ministry of Science and Technology of the People’s Republic of China National Key Research and Development Projects of China 2020YFD0900502 This study has been supported by the Central Public-interest Scientific Institution Basal Research Fund FMIRI of CAFS (NO. 2020YJS001), The Yangtze River basin ecological restoration engineering technology center of the Ministry of Agriculture (2021CG009), the Ministry of Science and Technology of the People’s Republic of China, and the National Key Research and Development Projects of China (2020YFD0900502). The funders had no role in study design, data collection and analysis, decision to publish, or preparation of the manuscript.

==============================
The hydrodynamic conditions of rivers affect fish habitats by influencing parameters such as river bottom topography. Ecological restoration projects change the water morphological characteristics of rivers. Here, water flow characteristics of the upper Yangtze River before and after the construction of a restoration project were analyzed using the computational fluid dynamics simulation method. The longitudinal diversion dam could divide the river into two flow velocity zones, and the outer flow is similar to the original river with a flow velocity of 0.75 m/s. However, flow velocity on the inner side of the river was about 0.25 m/s, forming a larger buffer area. The eddy became more diversified and stable, with a high eddy viscosity coefficient and less fluctuations, at 9 Pa·s; this was conducive to fish aggregation and spawning. At different depths, large gradient differences were observed between the inner and outer sides of the longitudinal diversion dam, and the turbulent current and upward flow of the inner side were obvious; this was more favorable to the aggregation of different fish species. The longitudinal dam body was under a pressure of about 200.2 Pa at the same flow rate; this was significantly lower than the pressure on the transverse dam body. The field flow test and fish survey data showed that the error rate of the simulation using the RNG turbulent model was less than 10% compared with actual mapping. After the restoration of fish habitats by the longitudinal diversion dam, the number of fish species in the area increased from 40 to 49; The density of fish in the water increased from 71.40 fish per 1,000 m2 before the project to 315.70 fish per 1,000 m2 after the project. These results can provide a reference for the rapid assessment of water morphology and fish habitat restoration in the future.

Introduction

In the Yangtze River Basin, large-scale construction facilities such as cascade hydropower, flood prevention, and bank protection facilities, bridges, and piers have been established. However, the morphological characteristics of downstream rivers were greatly altered, and the ecosystem has been adversely affected; this has caused endemic fish in the rivers to become endangered and seriously damaged the original biological chain of the rivers (Cooperative Group on fishery resources investigation of the Yangtze River system, 1990; Shi & Xu, 2008). River and bank zone remediation and restoration projects can improve the fish habitats in local river sections (Adeva-Bustos et al., 2019; Brown & Pasternack, 2010; Cooper & Knight, 2010). In addition, construction of narrow diversion dams in the river would have a positive effect on fish habitat quality, as they enlarge the buffer zone area and diversify the flow pattern (Cao et al., 2007; Zhao et al., 2019; Karaouzas, Theodoropoulos & Vourka, 2019). Thus, implementation of ecological restoration projects are essential for the restoration of fish habitats and protection of the aquatic ecosystem (Liu et al., 2011; Grizzetti, Pistocchi & Liquete, 2017; Lacey, Neary & Liao, 2012). For habitat restoration, appropriate assessment and quantitative evaluation of the advantages and disadvantages of fish habitats are necessary (Wu & Wu, 1994; Li, Hai & Fu, 2001; Ahmadi-Nedushan, St-Hilaire & Bérubé, 2010; Detenbeckn, Elonen & Taylor, 2010). Therefore, prediction models for all river fish habitat restoration projects are essential to avoid failure (Theodoropoulos, Stamou & Vardakas, 2020).

The fish life cycle is mainly composed of migration, growth, feeding, and reproduction. Considering the ecological significance of flow velocity gradient, its main function is to stimulate fish mating (Zhang et al., 2009; Anderson & Schwab, 2011; Anderson, Schwab & Lang, 2010). Most fish mainly reproduce by in vitro fertilization, and the flow with different gradient velocity is more favorable for mating and spawning. However, a high flow rate gradient can result in high shear stress and injure the fish species (Yi & Le, 2011; Wang, Dai & Dai, 2013; Bennion & Manny, 2011; Berg et al., 2013). Therefore, studying hydrodynamic factors that affect fish behaviour is a prerequisite for understanding the optimal habitat environment for fish species (Rong et al., 2012; Li et al., 2012; Brown & Pasternack, 2008). Flow rate is the most important environmental variable for fish habitat survival (Wang & Xia, 2010; Wu & Fu, 2007; Fu, Li & Jin, 2006). Therefore, hydrodynamic analyses of fish habitats are of great scientific importance for studies on ecological management and restoration of rivers and assessment of survival conditions of aquatic organisms (Zhang et al., 2021a; Fan et al., 2013; Austin & Wentzel, 2001; Yu, 2010).

In the world, hydrodynamic characteristics are generally used to evaluate the quality of fish habitats (Li, Lai & Li, 2020; Du et al., 2010; Wang, 2017; Yang, Yu & Chen, 2010). Yang, Yan & Qiao (2007) analyzed the hydrodynamic environment of fish habitats and showed that the fish species usually chose highly turbulent areas for mating behaviour and may stay in relatively calm waters to rest after spawning, indicating that areas with both turbulent and calm waters are used for fish spawning. Sun et al. (2015) used a two-dimensional river model to show the changes in habitat flow field, water depth and effective habitat areas before and after topographic remodeling of spawning grounds for Schizothorax prenanti. Sun, Zhang & David (2013) established a two-dimensional habitat model for the Heshui River, a backwater tributary, by coupling a hydrodynamic model and a fish information model for the habitats. The results showed that the fish habitats were distributed in a limited range of areas with a slow flow and complex habitat types on both banks and a large shallow habitat area with obvious eddy changes. In China, researchers have not only analyzed the relationship between fish behaviours and conventional hydraulic quantities but also established new characteristics to quantitatively describe the flow characteristics of fish habitats (Zhen, Wei & Liu, 2014; Chou & Chuang, 2010; Hardy, 1998; Bovee, 1982). Bai, Fang & He (2013) studied a three-dimensional large-eddy simulation of non-inundated dam bypass flow and showed that the ratio of dam length to the distance between dams has a significant impact on the flow form, turbulent flow intensity, and eddy distribution around fish habitats. Fu et al. (2016) performed a hydrodynamic simulation and fish habitat suitability simulation by using the in-channel flow increase method to understand the suitability of fish habitats in the Heshui River and proposed a plan for habitat restoration.

In contrast, other studies conducted abroad have principally simulated and analyzed the design parameters and effects of diversion dams in fish habitat restoration projects (Barmuta, 1990; Daneshvar, Nejadhashemi & Woznicki, 2017; Kolden, Fox & Bledsoe, 2015; Roni, Hanson & Beechie, 2008). Ettema & Muste (2004) conducted a series of flow pattern simulation tests on a model for diversion dams in a fish habitat and concluded that the shear stress parameter in the model was the main parameter that affected the return sideline of the dam and flow in the separation area. Ghodsian & Vaghefi (2009) placed a diversion dam at a 90° bend in the channel and concluded via simulation experiments that increasing the Froude number and length of the diversion dam would increase the scour volume of the channel and increasing the wing length of the diversion dam would reduce the scour volume and increase the diversion area. This provides a theoretical basis for restoration of fish habitats via engineering. Duan & Nanda (2006) also used a two-dimensional depth-averaged turbulent current model to simulate the relationship between the degree of scour and physical parameters in the area near a diversion dam.

Many studies on the hydrodynamic characteristics of fish habitats have been conducted in China and abroad (Zhang et al., 2013; Peller et al., 2006; Constantinescu, Sukhodolov & McCoy, 2009; Weitbrecht, 2004). However, few of these studies involve hydrodynamic simulations and field validation analyses of artificially restored fish habitats in the upper Yangtze River, especially turbulent and upwelling currents suitable for the survival of endemic fish. In this study, three different models were selected for simulation and analysis, and the hydrodynamic characteristics were compared. In addition, effects of diversion dams on the flow in front of and behind the dams and at different locations, flow velocity variation patterns on the cross-section and longitudinal section of the dam, and variation patterns of the turbulent flows and eddies generated by the different dams were analyzed. Lastly, results of the flow pattern test and fish resource survey after the ecological restoration project were compared to provide a reference for future ecological restoration projects in the Yangtze River channel.

Materials and Methods

Project area and monitoring time

Mituo Town river section is the central area of a rare and endemic natural fish reserve in the upper reaches of the Yangtze River in Luzhou City, Sichuan Province, and it is one of the few existing flowing river sections in the upper reaches of the Yangtze River after the first-cascade development of the Jinsha River and operation of the Three Gorges Project at full water level. Therefore, analysis of the river habitat in this section will be of great importance for fish reproduction, feeding, and habitat. The average river depth was about 5 m, and the average velocity was about 1–2 m/s. A straight tributary river of 800 m in length and 450 m in width was selected for the analysis; the average river bed gradient was about 1 in 10,000, with an average channel roughness of 0.015. The gradient change of the river was ignored during the experiment.

Topographic survey and mapping

The topographic survey and mapping were performed with the EchoSeep 300 multi-beam topographic sweeping equipment produced by LinkQuset, USA. The results showed that the average water depth was about 5 m, and the topography of the water bottom was relatively flat. A three-dimensional model of the river was established with the topographic data, and diversion dams, which were gravel soil mounds with a top height of 8 m and slope of 1:3, were added at different locations in the model for flow simulation. Then, flow patterns in case of no dams and transverse and longitudinal dams were analyzed.

Computational fluid dynamic simulations of the flow patterns and eddy morphology at an inlet velocity of 1 m/s were used to compare and analyze flow characteristics under different working conditions and select the most suitable solution for the construction of a diversion dam. Finally, the flow pattern after completion of the restoration project was tested using a moving ADCP flow velocity meter (LinkQuest, San Diego, CA, USA) and compared with the simulated data to verify the accuracy of the model.

Governing equations and turbulent model

Because the upper Yangtze River has a large area, turbulent currents, and many eddies, the RNG k−ε turbulent model and Reynolds-averaged Navier-Stokes equations were used. In this model, the influence of a small scale can be reflected by large-scale motions and modified viscosity terms, so that the small-scale motions are systematically removed from the governing equations. The model simultaneously considers the effects of eddies and low Reynolds number on turbulent current, which improves calculation accuracy in the presence of vortex flow; it is especially suitable for describing the complex flow environment, with shear flows having a large strain rate, eddy flows, and separation in the upper Yangtze River. The equations for turbulent kinetic energy k and turbulent dissipation rate ε are as follows:

(1) ρdkdt=∂∂xi[(αkμeff)∂k∂xi]+Gk+Gb−ρε−YM

(2) ρdεdt=∂∂xi[(αkμeff)∂ε∂xi]+C1εεk(Gk+C3εGb)−C2ερε2k−R

where Gk denotes the turbulent kinetic energy due to the mean velocity gradient, Gb denotes the turbulent kinetic energy due to the effect of buoyancy, and YM denotes the effect of compressible turbulence pulsation expansion on the total dissipation rate. αK and αε are reciprocals of the effective Prandtl number of turbulent kinetic energy k and dissipation rate ε, respectively. The model for calculating the turbulent viscosity coefficient is expressed as follows:

(3) d(ρ2kεμ)=1.72v~v~3−1−Cvdv~

where v~=μeff/μ. For integration of the previous equations, the effect of effective Reynolds number on turbulent transport can be precisely determined, which helps to deal with the simulation of a low Reynolds number and near-wall flow problems. For a high Reynolds number, the above-mentioned equation can yield: μt=ρCμk2ε, Cμ=0.0845.

Mesh generation and calculation

On the basis of the flow area and riverbed shape of the upper Yangtze River, a polyhedral mesh was used to divide the fluid area, which can greatly reduce the number of meshes and improve calculation efficiency. The mesh was locally encrypted near the irregular dams and in the middle of the river basin, where the flow changed dramatically. The mesh of the diversion dam model after division is shown in Fig. 1. Finally, Fluent software (Derek Dubner, New York, NY, USA) was used to find the solution, and the river bank, river bottom, and artificial dam parts in the flow field were defined as no-slip solid wall boundary conditions with a wall roughness of 0.03. The inlet velocity of the river was defined as a linear distribution that varied with water depth, with a velocity of 1 m/s at the highest point and bottom velocity of 0.2 m/s. The outlet of the flow field was defined as a pressure outlet boundary condition with a mixed fluid continuous phase density of 1,052 kg/m3 and kinematic viscosity coefficient of 1.0565 × 106 m2/s. The pressure-coupled algorithm was used to synchronize the iterations of velocity field and pressure field, and turbulent kinetic energy and turbulent dissipation rate were in the second-order up-wind format.

Figure 1 Schematic diagram of grid division: (A) no dam; (B) transverse dam; (C) longitudinal dam.

To compare the velocity on the basis of the experimental survey and mapping path, three states of the river were analyzed in this study: the pre-construction state without dam, the state with a transverse dam, and the state with a longitudinal dam. The right angle coordinate system was established using the oxy plane: x direction was the direction of water flow; y direction, direction of water width; and z direction, direction of water depth. In addition, QQ′, RR′, SS′, and TT′ were four transverse cross-sections parallel to the transverse dam body, and they were parallel to each other and 80 m apart. MM′, NN′, OO′, and PP′ were four longitudinal cross-sections of the water surface (Fig. 2).

Figure 2 Schematic diagram of section position under three different states.

Results

Simulation results for water flow state after engineering restoration

Characteristics of the flow pattern in the project area

Figure 3 shows velocity vector diagrams of the three models at the same incoming velocity. The results showed that, in the absence of the dam body, the entire flow field was relatively smooth, with a large flow velocity of about 0.8 m/s in the main channel and a few smaller eddies generated near the southern embankment. This was mainly due to the shape of the embankment. For the transverse dam model, the flow was blocked by the dam and suddenly decelerated to form an almost stationary low velocity zone and then separated to accelerate from both sides of the dam so that a path of high velocity of up to 1.0 m/s was formed between the dams. Because of the blockage by the dam, a low velocity zone was formed at the tail end of the dam so that small eddies and wake flows were evident. In the case of the longitudinal dam, the water flow was blocked by the dam and an obvious separation phenomenon was observed, in which the whole flow channel was divided into two distinctly different velocity areas; the flow on the outer side of the dam body was faster and stable, with a velocity of about 0.75 m/s. The velocity on the inner side of the dam was lower, around 0.25 m/s, and about one-third of the outer side, resulting in a buffer zone with an area about two-fifth of the entire channel. The fluid flowing through the barrier produced a reverse pressure gradient, forcing more fluid masses to stagnate and run backwards. As a result, large vortices were formed, with three large low-velocity vortex areas on the inner side of the dam near the river bank and a low-pressure area and many small vortices between dams. Foreign nutrients gathered here because of the field characteristics and material absorption by the eddies, would attract more fish.

Figure 3 Velocity of the flow field under different conditions.

Turbulent viscosity of water in the engineering zone

Turbulent viscosity refers to strong vortex cluster diffusion and cascade hashing due to random pulsations when the fluid flow is in a turbulent state, which looks like a fluid flow with great viscosity. The essence of turbulent viscosity is vortex diffusion, which is apparently understood as an increase in component viscosity (Wang, 2001). Previous studies have shown that stronger and more stable turbulent viscosity is more favorable for fish spawning (Wang & Tan, 2010). Turbulent viscosity at the cross-section is shown in Fig. 4.

Figure 4 Turbulent viscosity of different models.

(A) Cross-section of no dam; (B) cross-section of transverse dam; (C) cross-section of longitudinal dam; (D) longitudinal section of no dam; (E) longitudinal section of transverse dam; (F) longitudinal section of longitudinal dam. The x-axis of (A), (B), and (C) represent the location of the river channel, “0” is the middle of the river channel, negative value represents the inside of the river channel, and positive value represents the outside of the river channel.

The turbulent viscosity of the river in the three states was relatively smooth without the dam and around 1 Pa∙s. With the transverse dam, the entire flow field with eddies was mainly concentrated at the tail of the lower dam; the turbulent viscosity was relatively high, but the area was small; and the turbulent viscosity fluctuated between 1 and 8 Pa·s and was considerably unstable. With the longitudinal dam, turbulent viscosity near the dam body was relatively high, and the eddy was more stable; viscosity inside the dam body was significantly higher than that outside and was 9 Pa∙s. The results showed that the eddy generated by the transverse dam body was small and unstable; eddy viscosity was higher on the inner side of the longitudinal dam body and more stable over a large area, which was conducive to the aggregation of fish.

Flow velocity in different cross-sections of the dam body

Water velocity is an important parameter that reflects how a habitat is restored, and the effect of habitat restoration can be evaluated by simulating the effect of the dam on water flow distribution. Figure 5 exhibits the velocity distribution of the three models in transverse and longitudinal sections. The results showed that the main channel flow velocity was 0.75 m/s in the absence of the dam, and it slightly decreased in some areas near the southern bank. The transverse dam had a great impact on flow velocity, which slightly decreased at the front end of the dam, but increased sharply to 0.85 m/s after the flow separated and approached the middle area of the dam. After the flow passed through the longitudinal dam, the velocity in the tail section fluctuated sharply, with dramatic changes, and decreased and gradually stabilized as the flow moved away from the dam. The difference in velocity on the two sides of the longitudinal dam was large. The outer side of the dam was close to the main flow channel, where the velocity was higher and relatively stable, with a basic velocity of 0.75 m/s. The velocity on the inner side of the dam was lower, varying between 0.1 and 0.3 m/s, and a large buffer area was formed; the flow pattern was more complex, diverse, and conducive to fish foraging and aggregation, as it was affected by the backflow and circulation at the shore.

Figure 5 Velocity distribution of different models.

(A) Cross-section of no dam; (B) Cross-section of transverse dam; (C) Cross-section of longitudinal dam; (D) Longitudinal section of no dam; (E) Longitudinal section of transverse dam; (F) Longitudinal section of longitudinal dam; The x-axis of (A), (B), and (C) represent the location of the river channel, “0” is the middle of the river channel, negative value represents the inside of the river channel, and positive value represents the outside of the river channel.

Cross-sectional flow velocities of the dams at different water depths

Figure 6 shows average velocity distribution along the water depth direction for the three models at different cross-sections. The results showed few differences in velocity at water depth above 3 m on the same transversal, and the velocity decreased gradually from a water depth below 3 m to the river bottom. The velocity of the upper layer of the river was significantly greater than that of the lower layer. This was because the river bottom is a solid wall and a bit rough, which subjected the water flowing close to the river bottom to viscous resistance as well as frictional resistance of the river bottom. The addition of diversion dams without changing the velocity of the main channel can effectively reduce the velocity of water flow on the inner side of the dam, change the hydrodynamic environment of the fish habitat, and provide favorable survival conditions for different fish species and other aquatic organisms.

Figure 6 Velocity distribution along the depth direction of each section of different models.

(A) Cross-section of no dam; (B) Cross-section of transverse dam; (C) Cross-section of longitudinal dam; (D) Longitudinal section of no dam; (E) Longitudinal section of transverse dam; (F) Longitudinal section of longitudinal dam; “depth direction” indicates the direction below the water surface. “Depth position (m)” indicates the Depth below the water surface.

Water pressure on dam body

The service life of an engineering restoration project needs to be considered. In this study, two typical inflow dam construction locations were designed, so the pressure states subjected to the same flow velocity state needed to be analyzed. Figure 7 shows pressure distribution cloud images of the transverse and longitudinal dams subjected to the impact of water flow. The results showed that the maximum pressure on the transverse dams was 401.5 Pa, and each dam was subjected to a relatively high pressure. In contrast, the maximum pressure on the longitudinal dam was about 200.2 Pa; the pressure on the longitudinal dam was significantly lower than that on the transverse dam (P < 0.5). Therefore, the laterally arranged dams were more prone to damage and would have a relatively short life span.

Figure 7 Pressure distribution on the dams: (A) transverse dam; (B) longitudinal dam.

Validation of simulation results

To verify the reliability of the numerical simulation method and calculations, the flow velocity of the longitudinal dam body within the ecological engineering restoration area of the Luzhou section of Yangtze River was measured in September 2019 by using the ADCP Doppler flow velocity meter, and the flow velocity vector diagram are shown in Fig. 8. Table 1 shows the comparison between numerical simulation and test results at different positions.

Figure 8 The flow velocity vector diagram.

Table 1 Comparison of average simulated speed and test data.

Test line	Simulated value/(m·s−1)	Test value/(m·s−1)	Error analysis of flow velocities/%	
Line 1	0.75	0.78	4.00	
Line 2	0.65	0.70	7.69	
Line 3	0.28	0.26	7.14	
Line 4	0.22	0.24	9.09	

The test results showed that, after the longitudinal dam body was constructed, the flow in the test route outside the dam body at that time had an average velocity of 0.75 m/s, with a higher flow velocity, eastward direction, and stable flow pattern. The flow inside the dam had an average velocity of 0.25 m/s, greater change in direction, and more complex flow pattern. We compared the experimental test results and numerical calculation results and observed that the simulation results were basically consistent with the survey and mapping results and the error was less than 10%, indicating that the simulation of the flow pattern in the upper reaches of the Yangtze River was valid and accurate.

Discussion

Baril, Biron & Grant (2019) indicate that grid nodes with global habitat suitability scores N0.77 were considered suitable for lake sturgeon spawning by three-dimensional numerical simulation. Zhang et al. (2021b) analyzed the characteristics of fish convection field by in-situ observation and numerical simulation. Xia et al. (2017) use large eddy simulation (LES) and RNG k–ε calculate of three-dimensional flow field in the fishway by turbulence model, and compared and analyzed the flow field structures of typical sections at different times. However, they only used the numerical simulation method, and can not combine with the actual project. They did not evaluate the repair effect after the project. In this article, we compared results of the flow pattern test and fish resource survey after the ecological restoration project.

Before the implementation of the project, the average flow velocity was relatively fast in the study area, and the flow direction and gradient of flow velocity underwent small changes. After the implementation of the project, an obvious diversion phenomenon was detected when the water flowed through the diversion dam; several large vortex areas were formed inside the project, with the water flowing around rapidly and the shape of the vortex structure unchanged. The average flow velocity was low, flow direction changed faster, and changes in the gradient of flow velocity were larger. Therefore, the flow characteristics of the longitudinal dam body were more suitable for fish habitat reproduction than those before the construction of the diversion dam and those of the transverse dam body. To verify the findings of this study, the fish in the restoration area were surveyed and analyzed with drift net at an interval of 2 years before and after the construction of the diversion dam. The results showed that the number of fish in the dam increased significantly after the construction of the diversion dam, and the number of fish species in the dam increased from 40 to 49 (Fig. 9; Table 2). Previous studies have also shown that flow characteristics are key factors for fish habitats, and the spatial distribution of flow velocity reflects the complexity of water flow; different fish life stages and behaviours have compatible flow fields (Jason, Grzegorz & Laura, 2020). During reproduction, fish species with drifting eggs usually choose highly turbulent waters to mate, and mating behaviour is stimulated only when the flow reaches a certain level of turbulence. The fish species may stay in relatively calm waters before and after spawning (Yang, Yan & Qiao, 2007). For many fish species, reproduction requires complex signals from flow processes to stimulate gonadal maturation, ovulation, fertilization behaviours, and fish gonadal development requires sufficient dissolved oxygen. The flow rate is related to the amount of dissolved oxygen in the water (Baril et al., 2018). An area with a more complex flow pattern and eddies has a favorable aeration effect and high dissolved oxygen, which seems conducive to the development of fertilized eggs.

Figure 9 Comparison of fishes before and after project implementation.

Table 2 Fish survey data before and after project implementation.

Type	Before the project	After the project	
I ACIPENSERIFORMES			
1 Acipenseridae			
(1) Acipenser dabryarus Dumeril		◊	
II CYPRINIFORMES			
2 Cobitidac			
(2) Trilophysa bleekeri (Sauvage et Dabry)	⦾	◊	
(3) Misgurnus anguillicaudtus (Cantor)	⦾	◊	
3 Cyprinidae			
(4) Zacco platypus (Temminck et Schlegel)	⦾	◊	
(5) Opsariichthys bidens (Günther)	⦾	◊	
(6) Ctenopharyngodon idellus (Cuvier et Valenciennes)	⦾	◊	
(7) Distoechodon tumirostris (Peters)	⦾	◊	
(8) Xenocypris yunnanensis (Nichols)		◊	
(9) Xenocypris davidi (Bleeker)		◊	
(10) Hypophthmichthys mobitrix (Cuvier et Valenciennes)	⦾	◊	
(11) Rhodeus ocellatus (Kaer)	⦾	◊	
(12) Pseudolaubuca engraulis (Nichols)	⦾	◊	
(13) Pseudolaubuca sinensis (Bleeker)		◊	
(14) Sinibrama changi (Chang)	⦾	◊	
(15) Pseudobrama simoni (Bleeker)		◊	
(16) Hemiculter leucisculus (Basilewsky)	⦾	◊	
(17) Hemiculter tchangi(Fang)	⦾	◊	
(18) Hemiculterella sauvagei		◊	
(19) Erythroculter ilishaeformis (Bleeker)	⦾	◊	
(20) Erythroculter mongolicus mongolicus		◊	
(21) Ancherythroculter kurematsui (Sh. Kimura)		◊	
(22) Hemibarbus labeo (Pallas)	⦾	◊	
(23) Hemibarbus maculatus (Bleeker)	⦾	◊	
(24) Pseudorasbora parva (Temminck et Schlegel)	⦾	◊	
(25) Rhinogobio typus (Bleeker)	⦾	◊	
(26) Abbottina rivularis (Basilewsky)	⦾	◊	
(27) Abbottina obtusirostris	⦾	◊	
(28) Saurogobio dabryi (Bleeker)	⦾	◊	
(29) Squalidus argentatus		◊	
(30) Rhinogobio ventralis (Savage et Dabry)		◊	
(31) Acrossocheilus monticolus (Günther)	⦾		
(32) Cyprinus (Cyprinus) carpio (Linnaeus)	⦾	◊	
(33) Carassius auratus (Linnaeus)	⦾	◊	
(34) Procypris rabaudi (Tchang)		◊	
III SILURIFORMES			
4 Silurdae			
(35) Silurus asotus (Linnaeus)	⦾	◊	
(36) Silurus transverseis (Chen)	⦾	◊	
5 Bagridae			
(37) Pelteobagrus fulvidraco (Richardson)	⦾	◊	
(38) Pelteobagrus Vachelli (Richardson)	⦾	◊	
(39) Pelteobagrus nitidus (Sauvage et Dabry)	⦾	◊	
(40) Leiocassis longirostris (Günther)	⦾	◊	
(41) Leiocassis crassilabris (Günther)	⦾	◊	
(42) Pseudobagrus truncatus (Regan)	⦾	◊	
(43) Mystus macropterus (Bleeker)	⦾	◊	
6 Amblycipitidae			
(44) Liobagrus marginatus (Günther)	⦾	◊	
7 Ictaluridae			
(45) Ictalurus Punctatus (Rafinesque)	⦾	◊	
IV Synbgranchiformes			
8 Synbranchidae			
(46) Monopterus albus (Zuiew)	⦾		
VI PERCIFORMES			
9 Serranidae			
(47) Sinrperca kneri (Garman)	⦾	◊	
(48) Simiperca scherzeri (Steindachner)	⦾	◊	
10 Gobiidae			
(49) Ctenogobius giurinus (Rutter)	⦾	◊	
11 Channidae			
(50) Channa argus (Cantor)	⦾	◊	
12 Odontobutidae			
(51) Odontobutis obscurus (Temminck et Schlegel)	⦾	◊	

The fish density was about 71.40 fish per 1,000 m2 before restoration by the diversion dam. After project completion, a fish-echosounder was used to detect fish species on the inner side of the longitudinal dam. The results showed that the fish density on the inner side of the diversion dam was about 315.70 fish per 1,000 m2 after the construction; the fish density after project completion was about four times that before the restoration. This is mainly because the construction of the diversion dam changed the flow characteristics and produced more eddies, with significant changes in flow patterns and gradients, and allowed fish species to spawn in faster-flowing areas and rest and feed in still-water areas after spawning. Diversion dam construction provided better conditions for fish habitats in the upper reaches of the Yangtze River.

Fish habitats can be classified on the basis of their flow patterns, and different fish species have different habitat preferences. Therefore, habitat diversity is an important indicator of habitat quality. The construction of the diversion dam changed the topography of this river section, increased the total spatial area of the ecological niches of the fish community, and increased the vertical structure of the food web and fish trophic diversity. This resulted in a more uniform distribution of fish trophic ecological niches. Moreover, the flow characteristics changed and retention time of exogenous nutrients in the water increased, which could provide rich exogenous potential carbon sources and important energy recharge for the fish food web and a diverse habitat for fish species. The construction of the diversion dam divided the different ecological environments. This increased the number of eddies in the dam and made the flow pattern more complex, as the diversion dam contained both slow-flowing and fast-flowing areas as well as calm water areas. The increase in the area for trophic ecological niches for fish species living in slow flows increased the trophic path of the food structure for rare and endemic fish that inhabit the waters and had a positive effect on their growth, habitat, and baiting activities. Furthermore, increasing the proportion of omnivorous fish in the water (most of the fish are in the second trophic level) made the food web structure more complex, and a trophic redundancy was formed to maintain a stable fish food web structure in the area. The shallow water surface around the diversion dam, with strong light penetration and large phytoplankton biomass, led to water purification and significantly increased aquatic life-bearing capacity and biodiversity. This had a positive impact on fish reproduction in the gravel substrate of the river floodplains or in the deep waters of wide rapids. Therefore, the construction of longitudinal diversion dams had a positive effect on the restoration and improvement of fish habitats in the upper Yangtze River.

Conclusions

Simulations of the macroscopic motion of water flow in the artificially restored fish habitat in the upper Yangtze River were performed, and characteristics of the local eddy structure, flow velocity, and development and evolution of turbulent flow structure near the dam were studied. The data were compared with the flow velocity and eddies of the diversion dam before project implementation and at two typical locations. The following conclusions were obtained. 1. In this study, Fluent 3D hydrodynamics and habitat simulation software was used to compare and analyze flow pattern changes in the upper Yangtze River section before and after the construction of the diversion dam. Before project implementation, the flow velocity was faster, and the flow pattern was single. The transverse dam body could effectively slow down the flow velocity, but the eddy area was smaller; in contrast, the longitudinal dam body produced a more complex flow pattern with a large eddy area and part of the calm water area. This indicates that longitudinal arrangement of the diversion dams was more favorable to the recovery of fish resources in the upper Yangtze River.

2. The RNG turbulent model was used to simulate the flow pattern of the Yangtze River. After implementation of the longitudinal dam project, the turbulent flow velocity decreased, gradient changed dramatically, eddy flow increased, flow pattern became more complex, and backflow and circulation flow characteristics were obvious after the flow passed through the diversion dam. According to the field test, the error rate between the test data and simulation results was controlled to less than 10%. This indicates that the model can be better applied to flow simulation of the ecological restoration project in the upper reaches of the Yangtze River, reflects the flow pattern of the upper reaches of the Yangtze River fish habitat project, and describes the actual backflow generation and diffusion process around the diversion dam.

3. The analysis of fish species after the construction of the longitudinal dams showed that the fish species and density in the dams increased significantly. This indicates that the construction of longitudinal dams in the upper reaches of the Yangtze River was essential for the reconstruction and restoration of the fish ecosystem, improvement of the ecological function of the river, and maintenance of the stability of fish resources.

4. The analysis of flow patterns in the upper reaches of the Yangtze River and implementation of the project increased the total spatial area of the ecological niche of the fish community in the region. The water flow pattern had changed significantly, from a single flow field to a mixed flow field composed of upwelling current, back eddy, and stagnant flow, and provided a diverse habitat for fish species. Moreover, the water flow pattern affected the recovery of fishery resources. In particular, changes in the environment provided diverse habitats for rare and endemic fish species in the upper reaches of the Yangtze River and had a positive impact on their habitat improvement.

Supplemental Information

Supplemental Information 1 Fish survey data.

Fish resources survey before and after the project

Click here for additional data file.

Additional Information and Declarations

Competing Interests

Author Contributions

Data Availability

The authors are all members of Fishery Machinery and Instrument Research Institute, Chinese Academy of Fishery Sciences. The authors declare that they have no competing interests.

Xiaolong Chen conceived and designed the experiments, prepared figures and/or tables, authored or reviewed drafts of the article, and approved the final draft.

Xuan Che conceived and designed the experiments, authored or reviewed drafts of the article, and approved the final draft.

Xingguo Liu conceived and designed the experiments, authored or reviewed drafts of the article, and approved the final draft.

Lin Zhu performed the experiments, analyzed the data, prepared figures and/or tables, and approved the final draft.

Changfeng Tian performed the experiments, prepared figures and/or tables, and approved the final draft.

Xinfeng Li analyzed the data, prepared figures and/or tables, and approved the final draft.

The following information was supplied regarding data availability:

The raw data is available in the Supplemental Files.

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
