# Peer review of "Fish habitat restoration on the basis of water morphology simulation"

_PeerJ, doi:10.7717/peerj.13943_

## Round 0.1 · original submission · Major Revisions

Dear Authors, as you shall see that the reviewers have commented on your article and are suggesting significant revisions in the manuscript. Particularly regarding the improvements in each section pertaining to literature, coherence, and significance. In my opinion also discussion is weak. Use latest works and discuss your work's results in light of those relevant works.

While uploading your revised manuscript, please also upload the point-by-point response to both the reviewers.

Looking forward to receiving your revised manuscript soon.

Best,
Gowhar Meraj

·

Basic reporting

This is an interesting study and the authors have collected a unique dataset using cutting edge methodology. The paper is generally well written and structured. However, in my opinion the paper has some shortcomings in regards to some data analyses and text, and I feel this unique dataset has not been utilized to its full extent. In several instances I also suggested to cite more relevant and recent literature. Furthermore I made additional suggestions for more in-depth analyses of the data. Given these shortcomings the manuscript requires major revisions. Clear, explicit, professional English language has been used throughout the writing of paper. Structure conforms to PeerJ standards but may be improved for clarity. Figures are relevant, high quality, well labelled & described.
Abstract: Please focus the abstract on your study and your results. In particular the last two sentences are vague. I would prefer to see some data on turbulent viscosity of waters, Flow velocity in different cross-sections of the dam body from this study in the abstract, rather than a description of “where to go next”. More generally, I suggest focusing the manuscript on the scientific results rather than on the innovation in simulation.
Your introduction needs concise. I suggest that you improve the description to provide more justification for your study. The most important thing to include in the introduction of a research report is a clear, concise, explicit statement of the questions that the research was intended to answer. This statement should come near the beginning of the report —as soon as the reader has enough information to understand the reason for doing the research.

Experimental design

Include the important details from your research which are important enough to include in paper. If the study includes a questionnaire or a test that, the report should include examples of the questions or problems. It may also be worthwhile to include the entire questionnaire or test in an appendix, if it is not too long.

Validity of the findings

The overall findings are well stated linked to original research question and instrumental for future

Additional comments

The manuscript is clearly written in professional, unambiguous language.

Reviewer 2 ·

Basic reporting

Line 59-60, 71-73: The two expressions are almost identical. The authors should change the formation of the sentence.
Line 69-71: the sentence should be clearer.
Line 99-100: repetition of the “Heshui River”, one name can be deleted
Line 144: repetition of the “longitudinal”, one word should be deleted
Line 145: repetition of the “simulations”, one word should be deleted
Line 179: Is the repetition of the “New York” correct?
Line 266: no “longitudinal”, but “transverse” is correct
Line 268,269: the sentence is unclear “the pressure on the longitudinal dam was significantly lower than that on the longitudinal dam (P < 0.5).”
Line 281: “with a lower velocity “ maybe it can be deleted or changed?
Line 302 - 304: “During reproduction, fishes usually choose highly turbulent waters to mate, and mating behaviour is stimulated only when the flow reaches a certain level of turbulence.” What kind of fish species author is writing about? The main influence on many fish species reproduction is water temperature.
Line 303: correct “behavior” to “behaviour”
Line 306: Authors used “and” three times in one line in one sentence- better will be to delete or changed it
Line 34,35, 310, 313: My suggestion for authors is to think about what density scale will be the best. Perhaps it would be better to present results on fish density per 100 or 1000 m2? The maximum precision of this data presentation should be two decimal places.

In many sentences authors used “fishes”. Better definition is “fish species”.

Some sentences are too complex and too long. Authors should go through the text of the article again and split some sentences.

Literature from the last five years accounts for about 16% of the total literature used in the article. I suggest enriching the article with recent literature.

In the discussion authors focus on the advantages of longitudinal dam. In this section is only one citation. Author have to add more literature and compare the examinations and results with others articles in this topic.

Figure 1: Figures should be appropriately labelled, which is a, b and c. What is more, are the authors sure that the second image is a “transverse dam” and a third “longitudinal dam”?
Figure 3: If “nephogram” word is appropriately use in the description of this figure? Maybe better will be just to delete this word? The legend should be the same for the three images. In this case, one legend for three figures is sufficient. Numbers in the legend should be bigger, easy to read. Speed values are sufficient to two decimal places. The values should have a fixed graduation for example 0.00-0.25-0.50-0.75-1.00 etc.
Figure 4, 5, 6: The font in six images should be bigger, easy to read.
Figure 4abc,5abc: The graduation on the x-axis is unclear. Please add some explanation.
Figure 6: What does it mean “depth direction” and “Depth position (m)” ?
Figure 7: The scale of images should be the same. The size and the font of legends should be the same. The values on the legends should have a fixed graduation. For the first image for example 400, 250, 100, -50, -150 etc.

Experimental design

no comment

Validity of the findings

no comment

Additional comments

This submission presents a useful study for both fish habitats and fish community. In my opinion it should be improved. I am sure that after corrections it will be an interesting research for a wide range of scientists.

Annotated reviews are not available for download in order to protect the identity of reviewers who chose to remain anonymous.

·

Basic reporting

The resarch work of the paper is original in nature

Experimental design

Needs little modification with regard to Fish collection

Validity of the findings

Research paper is orginal in work and is great contribution towards River flow simulation models and helps greatly in knowing the effect of flow on fish species assemblages and occurence.

---

## Round 0.2 · accepted · Accept

Dear Authors,

Thank you very much for revising your manuscript. Overall the work has been considerably improved. I recommend its acceptance.
Best of luck, and serve humanity and planet Earth

·

Basic reporting

This is an interesting study and the authors have collected a unique dataset using cutting edge methodology.

Experimental design

Original primary research within Aims and Scope of the journal.

Validity of the findings

Valid findings

Additional comments

The paper is generally well written and structured.